# Investigation and Characterisation of New Eco-Friendly Cosmetic Ingredients Based on Probiotic Bacteria Ferment Filtrates in Combination with Alginite Mineral

Pál Tóth and Áron Németh *

Department of Applied Biotechnology and Food Science, Budapest University of Technology and Economics, Műegyetem rkp. 3., 1111 Budapest, Hungary
* Correspondence: naron@f-labor.mkt.bme.hu

**Abstract:** In light of the fact that three of the five healthiest meals on earth are fermented, fermentation came into focus of both customers, product developers, and researchers all around the world. Even in the cosmetics industry, fermented cosmetics have been increasingly introduced, creating a market emphasising the positive image that healthy fermented substances are environment-friendly and that it also aids skin health. Moreover, discovering usages for various naturally occurring organo-mineral rocks is a growing area of research. Thus, this study's aim was to combine the benefits of alginite and Lactobacilli (LAB) for cosmetic applications and investigate their combined effect on the skin considering the fermentation parameters as well, such as biomass and lactic acid concentration. The examined LAB strains were *Lactobacillus rhamnosus*, *Lactobacillus acidophilus*, *Limosilactobacillus reuteri*, and *Lactococcus lactis*, and a non-LAB probiotic strain *Bifidobacterium adolescentis* was also studied for the same purposes. The cell-free broth, also called as "filtrates", of the fermentations—both those that included alginite and those that did not—were tested for skin moisturising with a corneometer and for antioxidant activity with DPPH scavenging, as well as for skin-whitening properties with the inhibition of mushroom tyrosinase. The findings suggest that the combination of alginite and *Limosilactobacillus reuteri* is a potential novel cosmeceutical component with skin tanning capabilities. This result may help create more readily available, environmentally friendly, natural, and sustainable cosmetic ingredients.

**Keywords:** alginite; cosmetic; *Lactobacillus rhamnosus*; *Lactobacillus acidophilus*; *Limosilactobacillus reuteri*; *Lactococcus lactis*; *Bifidobacterium adolescentis*; fermentation



## 1. Introduction

The skin is the largest tissue that is thin and wide in humans, with a surface of about 2 m$^2$ and a mass of 4 kg [1]. The skin is mainly composed of the epidermis and dermis, separated by the junction of the two parts. Skin is essential to human health, as it is the body's first line of defence against harmful external physical, chemical, and biological invasions and other functions, such as controlling body temperature. In addition, skin is not only regarded as a physical barrier but also a dynamic tissue with its metabolism and the interaction between internal and external cells [1].

Figure 1 shows the skin's structure and the stratum corneum (SC) composition, including the natural moisturising factor (NMF). The skin consists of two main layers: the dermis and the epidermis. The epidermis is divided into two main layers: the viable epidermis and the stratum corneum (SC), the topmost layer mostly consisting of dead cells. The essential function of the SC is to act as a barrier, preventing dehydration resulting from the body's water loss. The SC contains 30% NMF, which consists of 40% amino acids such as serine, glycine, and alanine, and 12% lactate, which can retain water in the stratum corneum [2,3]. Due to this, several cosmetic products include lactic acid to restore lactic acid levels in the skin.

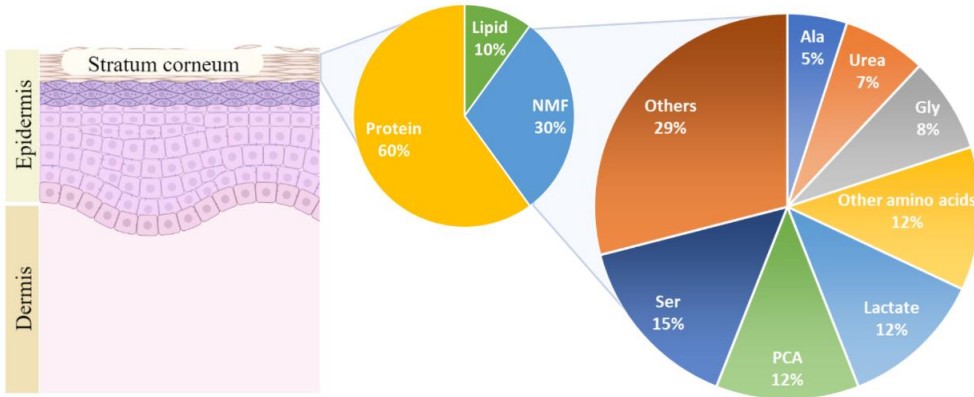

**Figure 1.** Composition of skin and natural moisturising factor. NMF natural moisturising factor Ala—alanine, Gly—glycine, Ser—Serine, PCA pyrrolidone carboxylic acid.

Lactic acid has many applications in the field of skincare. Lactic acid is used since ancient history when the famous Egyptian Queen Cleopatra was immersing her weekly milk bath, believing it would renew the skin and restore its youthfulness [4]. Beginning in the early 21st century, scientists at the Medical Center of the University of Maryland discovered that lactic acid plays a crucial and vital part in skin rejuvenation and skincare whenever people spend more than two hours each day doing so [5]. Studies have also shown that lactic acid plays a crucial role in skin whitening and the ergonomic arrangement of the melanin pigment, which determines the degree of skin colour. Treatment of brown spots or smudges with lactic acid at rates of 6–10% can help "directly inhibit the formation of melanin", resulting in the creation of an "enormous" tool to lessen or eliminate skin discolorations. [6]. Lactic acid peels exfoliate the skin's surface and help improve skin cell turnover rates. The removal of skin cells can help encourage the skin's production of natural elastin and collagen. This, in turn, may help reduce the appearance of wrinkles and fine lines [7,8].

Alginite is a non-core raw material from the fossilisation of accumulated organic (algae) and inorganic material, particularly clay, carbonates, quartz, and amorphous modification of silicic acid in the aqueous environment. Humic acids, as a component of alginite's organic portion, are known for their outstanding buffering ability, allowing them to stabilise pH throughout [9].

It has many benefits and usages in agriculture because of its unique composition and water adsorption capability. Numerous research on this mineral has demonstrated that, depending on the dose, alginite may be used to significantly boost agricultural productivity [9–11]. Nowadays, there are some recent studies about innovative approaches to the usage of alginite. For instance, in 2017, Hlubeňová, K., and colleagues compared the effect of the mineral in specified pathogen-free (SPF) mice with and without infections by *Salmonella typhimurium* CCM 7205 on its own (Alg, Alg + Sal) and after the mineral's fermentation by *L. reuteri* 2/6 (Lab + Alg, Lab + Alg + Sal) [12]. The results were compared with data from mice that received neither alginite nor *L. reuteri* fermentum. Their experiment indicated an immunomodulation potential of lactobacilli and humic substances found in alginite. In the group infected with *S. Typhimurium*, the cellular immune response was significantly stimulated by the administration of *L. reuteri* and alginite, particularly at the local level, in mesenteric lymph nodes of mice by activation of CD4+ CD8+ lymphocytes, NK (Natural Killer) and NKT (Natural Killer T-cells) cells, and at the level of the innate immune system component by the activation of phagocytosis.

Additionally, in 2017, during another research study [13], alginite was added at a dose of 1% for 14 days to the canine diet along with probiotic supplementation, and this resulted in a reduction of coliform and clostridium-like bacteria and an increase in lactic acid bacteria, as well as an increase in haemoglobin concentration, stimulation of cellular immunity parameters, and an improvement in the serum mineral levels. Both the consistency of

the faeces and the appetite remained unchanged. Therefore, it is appropriate to combine alginite and *L. fermentum* CCM 7421; however, using alginite alone had some unfavourable outcomes, including an increase in the population of clostridium-like bacteria, a decrease in haemoglobin concentration, or an increase in the levels of alanine aminotransferase.

In today's modern world, using natural ingredients in cosmetics has come to the fore again [14–19]. The EU cosmetic ingredient database contains more than 1000 fermented ingredients. However, only a few scientific reports can be found on their production and application. Most of these new cosmetic ingredients are made with lactobacillus strains [20]. Our laboratory has been dealing with probiotic lactic acid producers for many years [21–24], but so far, we have not performed any measurements in the cosmetic direction. The benefit of alginite and its synergistic effect with lactic acid bacteria described gastroenterally inspired us to investigate the effects of alginite in conjunction with lactic acid-producing bacteria and other probiotics on skin for a potential cosmetic application.

## 2. Materials and Methods

### 2.1. The Fermentation

The strains were purchased from the National Collection of Agricultural and Industrial Microorganisms and the National Collection of Industrial, Food and Marine Bacteria. The following microorganisms and mediums were used in this study: *Limosilactobacillus reuteri* (NCIMB 11951) in MRS, *Lactobacillus rhamnosus* (NCAIM B.02274) in MRS, *Lactobacillus acidophilus* (NCAIM B.02085) in MRS, *Lactococcus lactis* (NCAIM B.02123) in M17, *Bifidobacterium adolescentis* (NCAIM B.01822) in Bifidobacterium medium.

De Man, Rogosa and Sharp (MRS) medium contains the following: Peptone 10 g/L; Meat extract 10 g/L; yeast extract 5.0 g/L; Glucose 20 g/L; $K_2HPO_4$ 2.0 g/L; Sodium-acetate 2.0 g/L; Ammonium-citrate 2.0 g/L; $MgSO_4 \times 7H_2O$ 0.2 g/L; $MnSO_4 \times H_2O$ 0.05 g/L; Tween-80 1.08 g/L.

M17 medium: Tryptone 5.0 g/L; Soy peptone 5.0 g/L; Beef extract 5.0 g/L; Yeast extract 2.5 g/L; L-Ascorbic acid 0.5 g/L; Magnesium-sulphate 0.25 g/L; Disodium glycerol-β-phosphate 10 g/L and Lactose 5 g/L.

Bifidobacterium medium: Peptone from casein 10 g/L; Yeast extract 5.0 g/L; Meat extract 5.0 g/L; Soy peptone 5.0 g/L; Glucose 10 g/L; $K_2HPO_4$ 2.0 g/L; $MgSO_4 \times 7 H_2O$ 0.2 g/L; $MnSO_4 \times H_2O$ 0.05 g/L; Tween-80 1.0 mL; Sodium chloride 5.0 g/L; Cystein-HCl $\times H_2O$ 0.5 g/L; Resazurin (25 mg/100 mL) 4.0 mL; Trace elements solution 40 mL.

Trace elements solution: 1000 mL distilled water: $CaCl_2 \times \cdot 2H_2O$ 0.25 g; $MgSO_4 \times 7H_2O$ 0.50 g; $K_2HPO_4$ 1.00 g; $KH_2PO_4$ 1.00 g; $NaHCO_3$ 10.00 g; NaCl 2.00 g.

In the case of the alginite-based fermentation, each medium was supplemented with powdered alginate mineral (Gérce, Hungary) 10.0 g/L if it was necessary.

The fermentations were carried out in a 1 L benchtop bioreactor with a working volume of 0.8 L (Biostat Q fermenter, B. Braun Biotech International, Melsungen, Germany) and a 5% *v/v* inoculum. For production, the temperature was adjusted to 37 °C with an agitation speed of 300 rpm. The pH was controlled by 25% $H_3PO_4$ and 25% NaOH.

After the fermentations, each broth was centrifuged, and the supernatant and cells (with alginite) were separated.

### 2.2. Skin Moisturising Measurement

The ferment filtrates' short-term/immediate hydration effect was determined using a dermatoscope, as we previously reported [25]. We marked a one-square-centimetre area on the forearm three times and pipetted 20 microliters of cell-free ferment filtrate. After 5 min, we wiped it with a dry hand towel and then measured the hydration of that part of our skin at given intervals with the Corneometer (capacitive) sensor of Multi Dermascope MDS 800. In order to have a basis for comparison, we measured the level of hydration in the skin before the measurement and subtracted that value from each measured value.

## 2.3. Antioxidant Capacity Measurement

The ferment filtrates antioxidant capacity was determined with the 1,1-Diphenyl-2-picrylhydrazyl (DPPH, 97%) scavenging method [26]. For the calibration L-ascorbic acid (AscH2, 99,82%) was used in UV/HPLC grade methanol which were purchased from Sigma Aldrich Chemical Co. (St. Louis, MO, USA). Fresh stock solutions were prepared before each analysis. The spectrophotometric measurements were performed in a Pharmacia LKB-Ultrospec Plus spectrophotometer with 1 cm glass cuvettes at 517 nm.

For DPPH·activity measurement, a 150 µmol/L methanolic solution was prepared. For DPPH-H, the methanolic solution was prepared with DPPH· and AscH2, both at 150 µmol/L (100% excess of AscH2), protecting the reaction from light for 0.5 h. The DPPH assay was performed by adding constant aliquots of 1.5 mL of an AscH2 methanolic solution (300, 150, 75.0, 37.5 and 18.75 mmol/L) to 1.5 mL of a DPPH solution (150 µmol/L in methanol). The same procedure was applied to the ferment filtrates, although there were some modifications. The fermentation filtrates were diluted twice with methanol, and the solutions were kept in the fridge for a night. The precipitated substances were centrifuged, and the separated supernatants were used hereafter in different fold dilutions (2, 4, 8, 16 and 32). The negative control was prepared by 1.5 mL methanol and and 1.5 mL 150 µmol/L DPPH in methanol. All the reactions were kept in the dark for 30 min until measurements. All samples and negative controls were measured in triplicates. The following equation calculated the percentage DPPH scavenging activity ($Abs_{NC}$—average absorbance of negative control, $Abs_{samp}$—average absorbance of sample):

$$DPPH_{Scav.}(\%) = \left[ \frac{(Abs_{NC} - Abs_{samp})}{Abs_{NC}} \right] * 100 \tag{1}$$

## 2.4. Mushroom Tyrosinase Inhibition

The ferments' filtrates' tyrosinase inhibitory activity was evaluated to determine the filtrates' skin-whitening activity using mushroom tyrosinase and L-DOPA (98%). The method reported by Toshiya M. [27] was employed with some modifications. Briefly, the cuvettes were designated for A (negative control), B (blank of negative control), C (sample), and D (blank of the sample), which contained the following reaction mixtures: A, 750 µL of a 1/15 M phosphate buffer (pH 6.8) and 200 µL L-DOPA (10 mM in the same buffer with 5% DMSO); B, 950 µL of the same buffer; C, 350 µL of the same buffer, 200 µL L-DOPA (in the same solution), and 400 µL of an appropriate amount of the sample; D, 550 µL of the same buffer and 400 µL of the same amount of the sample solution. B and C cuvettes for all samples were determined in triplicates. The contents of each were well mixed, and 50 µL of tyrosinase (50 units/mL in the same buffer) was added. After incubation at room temperature (23 °C) for 10 min, each cuvette's absorbance at 475 nm was measured in Pharmacia LKB-Ultrospec Plus spectrophotometer. The following equation calculated the percentage inhibition of the tyrosinase activity:

$$tyrosinase\ inhibiton\ (\%) = \frac{[A - B] - [C - D]}{[A - B]} \times 100 \tag{2}$$

The reference compound used for calibration was kojic acid.

## 3. Results

### 3.1. Skin Moisture Effect Determination

First of all, we determined the ferments' filtrates' moisturising effect on human skin. As all of the used probiotic strains' main product is lactic acid, we initially assessed the moisturising impact of lactic acid solutions at various concentrations. Furthermore, the following concentrations were used: 5, 10, 15, and 20 g/L of lactic acid in distilled water. After applying a drop of the lactic acid solution with the desired concentration, the skin's hydration varies over time, as shown in Figure 2. It is clear that, as time passes, the

hydration value rapidly drops but eventually stabilises at different relative moisture values (after 35 min), according to the lactic acid (and other NMF) concentration.

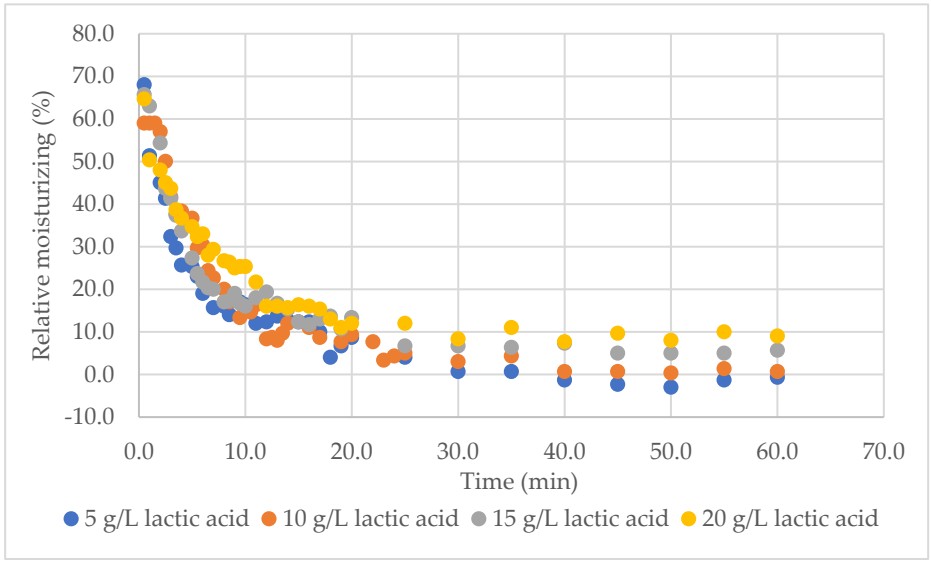

**Figure 2.** Moisturising effects of lactic acid solutions versus time.

The results obtained in the 35th, 40th, 45th, and 50th minutes were averaged and plotted against the lactic acid concentration (Figure 3). A trendline was fitted to the obtained points, which we were able to use to predict the hydrating effect of the fermentation filtrates according to their lactic acid concentrations.

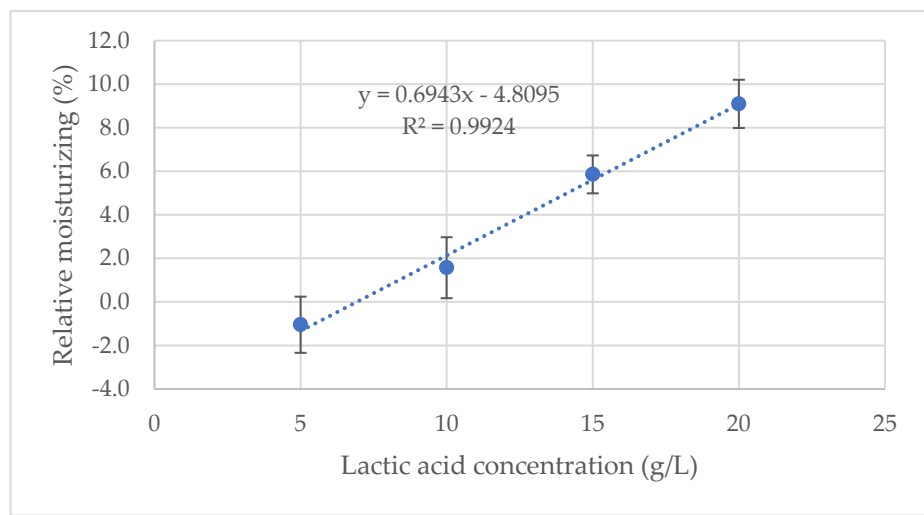

**Figure 3.** The skin moisturising of different lactic acid solutions.

The moisturising effect calculated from the lactic acids produced can be seen in Table 1. The calculated and measured values do not differ significantly, with the exception of *L. lactis*, which resulted in a negative value. However, as is the case with other humectants, the application of lactic acid alone fails to ameliorate the symptoms of dry skin and coformulation with occlusive agents is required to help retain the humectant-bound water within the surface layers of the SC. Typically, researchers have found that lotions containing barrier lipids (ceramides) and lactic acid provide a synergistic relief from dry skin [28].

**Table 1.** The calculated moisturising effect of ferment filtrates according to their lactic acid concentration with the measured results.

| | Lactic Acid Conc. (g/L) | | Calculated Moisturising Effect (%) | | Mean of Measured Moisturising Effect (%) | | |
|---|---|---|---|---|---|---|---|
| | **with A.** | **without A.** | **with A.** | **without A.** | **with A.** | **without A.** | *p* * |
| *B. adolescentis* | 14.6 | 14.6 | 5.33 | 5.33 | 4.00 ± 0.71 | 8.89 ± 2.86 | 0.000138 |
| *L. lactis* | 5.0 | 5.5 | −1.34 | −0.99 | 8.78 ± 0.73 | 9.00 ± 1.12 | 0.623876 |
| *L. reuteri* | 17.0 | 19.2 | 6.99 | 8.52 | 7.89 ± 2.49 | 9.11 ± 2.32 | 0.296743 |
| *L. rhamnosus* | 18.4 | 22.0 | 7.97 | 10.47 | 9.78 ± 2.05 | 9.78 ± 1.01 | 0.999997 |
| *L. acidophilus* | 20.0 | 18.7 | 9.08 | 8.17 | 5.67 ± 0.65 | 12.08 ± 0.79 | 0.000000 |

* *p* < 0.05 was considered indicative of significance (*t*-test).

The column diagram below, with one alginite (with A.) and one non-alginite (without) per strain, illustrates the hydrating impact of the filtrates (Figure 4).

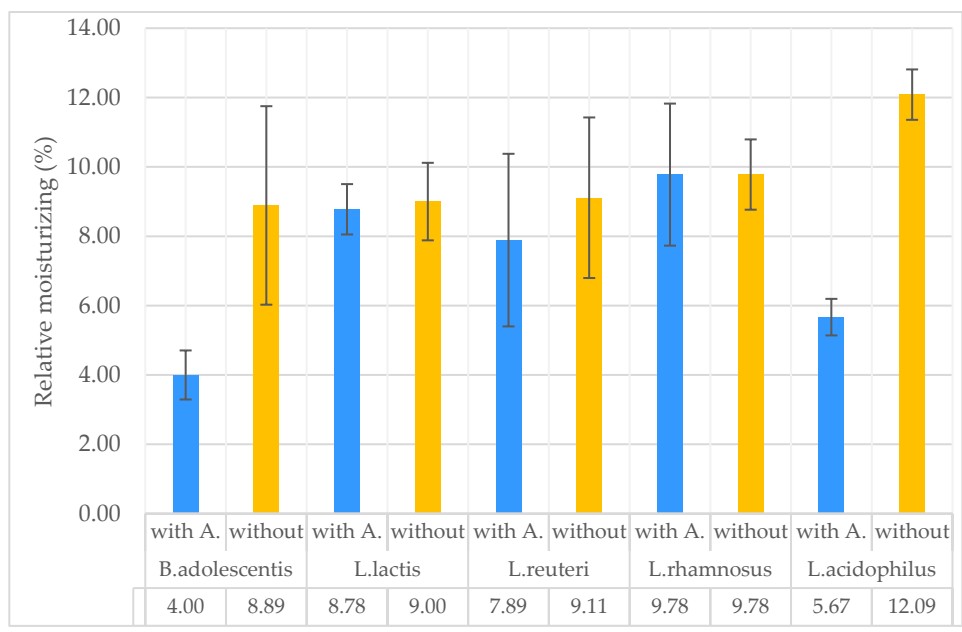

**Figure 4.** The ferment filtrates rel. moisturising in a steady state. (with A. = fermentation media included alginite and without = the fermentation media did not include alginite).

Interestingly, the vast majority of the relative moisturizing results are between 8 and 10%, and lower values are found only in two cases: *B. adolescentis* and *L. acidophilus*, both in combination with alginite. In these two cases, the relative moisturizing of the filtrates of fermentations without alginite and with alginite differ significantly (Table 1). It should be noted that this does not mean that the alginite LAB fermentation filtrate dries the skin, since we still found a higher value than the hydration measured before the measurements. The fact that alginite did not cause drying, despite its high mineral and carbonate content, is encouraging.

*3.2. The Antioxidant Capacity*

There is an increasing demand for the quantitative determination of antioxidant capacity (AOC), which is why many analytical procedures and measurement systems have been developed in the past decades. AOC measurement methods can be divided into two main groups. The first group is based on an electron transition, such as (e.g., 2,2-diphenyl-2-picrylhydrazyl (DPPH) radical scavenging; ferric-reducing antioxidant power (FRAP), and cupric reducing antioxidant activity (CUPRAC)) measurements, while the second group

includes methods depending on a hydrogen atom transfer (e.g., oxygen radical absorbance capacity (ORAC), photo chemiluminescence (PCL), and total radical-trapping antioxidant potential (TRAP)) [29].

DPPH scavenging results are shown in Table 2, Similarly to the results of [30]. IC50 values represent the amount of the filtrates, which reached 50% inhibition, so the lower the value, the better the result is. In all cases, with one exception (*L. acidophilus*), the presence of alginite reduced the antioxidant effect, which was surprising, as it was previously reported that alginite contains humic and fulvic acids, which are known to have excellent antioxidant effects [31–35]. However, we did not particularly expect the effect of humic acids, since they can only dissolve in an alkaline medium. In the case of *L. lactis*, we got much better results than all the others because the medium contains L-ascorbic acid, which is a perfect antioxidant.

**Table 2.** The DPPH scavenging results of ferment filtrates.

| Sample | Regression Equation | IC$_{50}$ |
|---|---|---|
| *B. adolescentis* with. A. | f = $-8.1871 \times$ x + 1 (R$^2$ = 0.9995) | 6.1% |
| *B. adolescentis* without | f = $-8.9354 \times$ x + 1 (R$^2$ = 0.9985) | 5.6% |
| *L. lactis* with. A. | f = $-15.508 \times$ x + 1 (R$^2$ = 0.9984) | 3.2% |
| *L. lactis* without | f = $-16.942 \times$ x + 1 (R$^2$ = 0.9998) | 3.0% |
| *L. reuteri* with. A. | f = $-4.618 \times$ x + 1 (R$^2$ = 0.9952) | 10.8% |
| *L. reuteri* without | f = $-5.3366 \times$ x + 1 (R$^2$ = 0.9923) | 9.4% |
| *L. rhamnosus* with. A. | f = $-3.9569 \times$ x + 1 (R$^2$ = 0.9993) | 12.6% |
| *L. rhamnosus* without | f = $-4.223 \times$ x + 1 (R$^2$ = 0.9957) | 11.8% |
| *L. acidophilus* with. A. | f = $-6.5685 \times$ x + 1 (R$^2$ = 0.9976) | 7.6% |
| *L. acidophilus* without | f = $-6.2333 \times$ x + 1 (R$^2$ = 0.9885) | 8.0% |
| L-ascorbic acid | f = $-0.128 \times$ x + 1 (R$^2$ = 0.9988) | 39.0 µmol/L |

### 3.3. Mushroom Tyrosinase Inhibition

Tyrosinase inhibition is primarily intended to identify sunscreens and their effectiveness in preventing skin disorders caused by melanin overproduction, such as melasma and *melanoma malignum* [36]. Tyrosinase is involved in melanin anabolism in melanocytes. This enzyme catalysis two different reactions: the hydroxylation of monophenolic compounds to o-diphenols and the oxidation of o-diphenols to o-quinones. The enzyme converts tyrosine to 3,4-dihydroxyphenylalanine (DOPA) and oxidises L-DOPA to form dopaquinone, which plays a role in melanin biosynthesis.

The IC$_{50}$ value is commonly used for quantitative comparison, indicating that effective concentration causes enzyme inhibition of 50%. However, among the used five bacteria, only *Bifidobacterium adolescentis*, *Lactococcus lactis*, and *Limosilactobacillus reuteri* (Figure 5a) reached the 50% inhibition as a threshold, and the other two strains had only a slight inhibitory effect. On the other hand, all alginite fermentation inhibited enzyme inhibition and even boosted it in some cases. For instance, *L. lactis* and *L. rhamnosus* (Figure 5b), alginite and alginite-free ferment filtrate had a tyrosinase inhibitory effect, although alginite reduced the level of inhibition in both strains. Therefore, for the sake of simplicity, we evaluated the maximum values achieved.

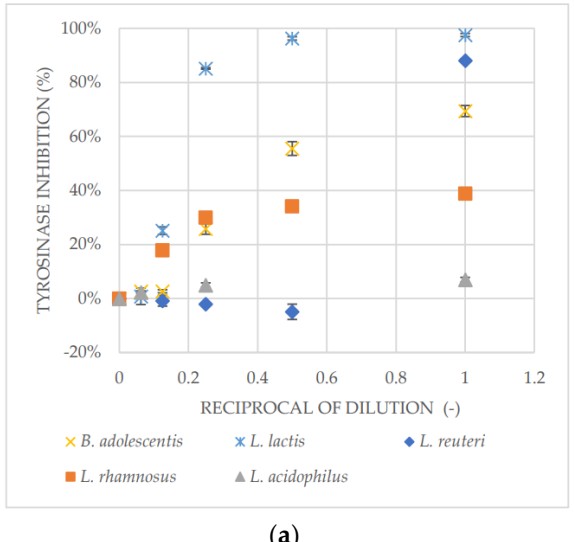

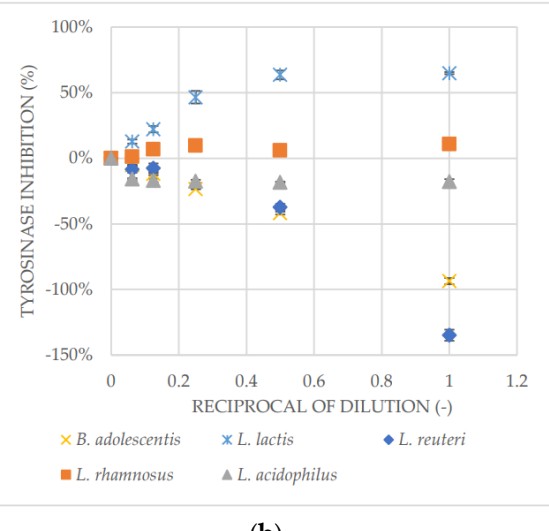

(**a**)                                             (**b**)

**Figure 5.** The dilution rates' reciprocal of tyrosinase inhibition of probiotics ferment filtrates: (**a**) alginite-free ferment filtrates and (**b**) alginite-based ferment filtrates.

The highest achieved inhibition of tyrosinase *L. reuteri* and *B. adolescentis* alginite-free filtrates had a value of 88% and 69%, respectively. However, the alginite-based ferment filtrates for the same strains reached values of −94% and −135% at the same dilution.

## 4. Discussion

We conducted experiments to reveal a new area of use for alginite, namely within the cosmetics industry. As in previous reports, we combined alginite with five probiotic (mainly lactic acid-producing) bacteria: alginite and LABs had positive synergetic effects. Therefore, we examined the fermentation filtrates with and without alginite and evaluated the preparations based on the following: skin hydration effect, antioxidant capacity (AOC), and skin-whitening effect.

According to voluntary skin hydration measurements, alginite decreased water retention in the skin compared to alginite-free ferment filtrates. However, the alginite-based ones still had a positive moisturising effect in all cases. Furthermore, the differences between alginite-supplemented and alginite-free filtrates were found insignificant for three of the five strains. Furthermore, lactic acid also has a moisturising activity, and a large amount of calcium carbonate in alginite helps deactivate this compound by precipitating it in form of calcium lactate.

As a result of determining the antioxidant capacity, we found that alginite significantly reduced the antioxidant effect in all cases, except for *Lactobacillus acidophilus*. The reason for this may be because the humic and fulvic acids were not able to dissolve from the alginite into the acidic medium. Additionally, the lack of a readily oxidisable group in fulvic acids (Figure 6) may be a sign that they are weak or not antioxidant. Moreover, lactic acid also has antioxidant activity, and the large amount of calcium carbonate in alginite helps deactivate this compound by precipitating it in the form of calcium lactate, similarly to a moisturizing effect. Taking these details into account, it is easy to understand why our ferments without alginite had better results. However, DPPH scavenging measurement is a well-established method for determining the antioxidant effect, even though the experiment is typically performed in methanol or ethanol. The addition of both solvents to the fermentation filtrate precipitates some compounds. In this consciousness, the measurement can only determine the existence and effect of an antioxidant substance(s) that dissolves in these two solvents.

(**a**)  (**b**)

**Figure 6.** Organic humin-based chemical compounds are found in alginite: (**a**) the general chemical structure of fulvic acid; (**b**) humic acids.

In the study of mushroom tyrosinase inhibition, we obtained unexpected results because humic and fulvic acids can bind metal ions, and alginite-supplemented ferment filtrate should also complex copper ions, which are essential for the functioning of the tyrosinase enzyme. In contrast, alginite-containing filtrates were found to reduce the inhibitory effect of tyrosinase, i.e., activate the enzyme. Consequently, alginite-free filtrates act as an inhibitor of tyrosinase, whereas alginite-based filtrates act as a booster, especially in the case of *L. reuteri* and *B. adolescentis*.

The results may seem surprising initially, since humic and fulvic acids (Figure 6) contain many carboxyl and hydroxyl groups at a sufficient distance to bind copper ions similarly to kojic acid, which ions are essential for the enzyme to function while producing precursor for melanin synthesis.

Humic acid has been reported to react with calcium oxalate and complex calcium ions [37]. Humic acid can probably bind copper ions similarly (Figure 7).

**Figure 7.** Illustration of the humous acid's interaction with copper ions using the structural fragment of humous acid.

On the other hand, alginite also has many metal ions [38] that contribute to the functioning of the enzyme, such as $Co^{2+}$, $Zn^{2+}$ and $Cu^{2+}$ [39,40]. According to the presented results, instead of inhibition by alginate through removal of necessary ions, alginite rather released metal ions into the assay mixture, resulting in enhanced tyrosinase activity.

## 5. Conclusions

The alginite-enriched ferments of these two bacteria (*Limosilactobacillus reuteri* and *Bifidobacterium adolescentis*), after being tested against skin toxicity, might be mixed into an ideal formula and with the appropriate dosage instructions to produce an excellent tanning cream. Furthermore, the presented fermentation-based cosmetic ingredients are made from sustainable and environmentally friendly materials; meanwhile, they are highly effective.

**Author Contributions:** Conceptualization, supervising, funding acquisition: Á.N., methodology, validation, formal analysis, original draft preparation: P.T. All authors have read and agreed to the published version of the manuscript.

**Funding:** The APC was funded by MÉL Biotech K+F Kft. (Budapest, Hungary).

**Informed Consent Statement:** Informed consent was obtained from all subjects involved in the study.

**Data Availability Statement:** Supporting data are available on request from corresponding author.

**Acknowledgments:** The authors are grateful for Alginit Ltd. (Gérce, Hungary) providing the alginite mineral and for MÉL Biotech K + F Kft (Agro-, Food Biotechnology R&D Ltd., Budapest, Hungary) for the dermatoscope.

**Conflicts of Interest:** The authors declare no conflict of interest.

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
