# Peer review of "Investigation and Characterisation of New Eco-Friendly Cosmetic Ingredients Based on Probiotic Bacteria Ferment Filtrates in Combination with Alginite Mineral"

_processes, doi:10.3390/pr10122672_

Round 1
Reviewer 1 Report
The manuscript on ‘Production and characterization of new eco-friendly………………….’ is quiet interesting approach to study the filtrate of lactic acid producing -probiotic with or without alginate as skin moisturizing effect and tyrosinase inhibitor.
However, authors need to address the major quires as follows
1. Did the authors quantify the lactic acid production in the filtrate of bacterial cultures with or without alginate ?.
2. From the results, we understood that alginate has not direct synergistic effect of alginate with lactic acid. Please clarify
3. How tyrosinase inhibition activity related to skin moisturizing effect ?. If so include the relationship in the discussion section.
4. As metal ions involved in tyrosinase activity, clarify have the authors tested with metal ions individually.
5. Perform statistical analysis for the experimental dats
Minor corrections
1. There are typo errors ex. 0,5 h, 1,5 ml instead 0.5h and 1.5 ml. Please check throughout the manuscript
2. Typo errors in line number 223, 226
Author Response
The authors are very grateful for receiving the reviewer's comments and suggestions, to which they give the following answers:
- Did the authors quantify the lactic acid production in the filtrate of bacterial cultures with or without alginate ?.
Yes, we did, and Table 1. indicates these results, which are now also considered in the renewed Discussion paragraph.
- From the results, we understood that alginate has not direct synergistic effect of alginate with lactic acid. Please clarify
According to our skin-tests, alginite (not alginate) had no synergistic effect with lactic acid containing ferment filtrates. Manuscript was supplemented (Line 291-293)
- How tyrosinase inhibition activity related to skin moisturizing effect ?. If so include the relationship in the discussion section.
No, tyrosinase inhibition activity is not related to skin moisturising, it is another different skin-test for cosmetic ingredients.
- As metal ions involved in tyrosinase activity, clarify have the authors tested with metal ions individually.
It was already tested by several authors, which are also cited in the manuscript [38-39], therfore we did not test.
- Perform statistical analysis for the experimental dats
Representation of DPPH results was changed from Figure to Table, where regression equations and their regression coefficients are added as statistical indicator. Also Error bars were added to Figure 5.
Minor corrections
- There are typo errors ex. 0,5 h, 1,5 ml instead 0.5h and 1.5 ml. Please check throughout the manuscript
Comas were changed to decimal dots along the whole manuscripts according tot he above suggestion.
- Typo errors in line number 223, 226
Typos were corrected throughout the whole manuscript, including two samples of „alginate” to „alginite”, since the latter was used.
Reviewer 2 Report
The title needs to be modified for better understanding. The English of the Ms. needs to be checked by professional services and some of the sentance needs to be reframed i.e. Since three of the top five healthiest meals on the planet have been found to be fermented, fermentation is a topic of interest to people all around the world. In figure 1, the abbreviated forms need to be explained in the legend. Line no. 122 the details of Alginate grade, type and make is required. Authors need to add some details such as FTIR, or MS data of culture filtrate to understand the end products of the fermentation.
Author Response
The authors are very grateful for receiving the reviewer's comments and suggestions, to which they give the following answers:
A thorough lingual improvement was done along the whole manuscript rephrasing many sentences, the first one become as follows: In light of the fact that three of the five healthiest meals on earth are fermented, fermentation came into focus of both customers, product developers and researchers all around the world.
We suggest the following modified Title:
“Investigation and characterisation of new eco-friendly cosmetic ingredients based on probiotic bacteria ferment filtrates in combination with alginite mineral”
Abbreviations in Fig.1. are now solved in figure caption.
In Line 122 (now Line 128) alginite was specified as “powdered alginate mineral (Gérce, Hungary)” beside the modification from alginate to alginate.
The manufactured and investigated "product" of the fermentation was the broth itself, which is a mixture of compunds. Its spectra (both FTIR and MS) is varying with media composition and fermentation process, but its effect is more characterizing. At the same time, this is a first report on usefullness of the complete broth for skin improvement, we intended to verify its benefit, but later on we are going into more details and identify particular active molecules probably using FTIR and/or MS.
Reviewer 3 Report
This is a well presented work on the use of the culture free supernatant form probiotics grown in the presence of alginite on skin care products. However, the results were negative towards the intended target, which was moisturization, and the authors recommend their product for self tanning purposes. Such a product should be further evaluated for skin irritations on long term use. I would also like to draw your attention to the following:
Introduction: The first sentence is too general. Your target is the skin and it might be appropriate to elaborate on the existing literature. Please provide relevant references.
Introduction p 3, L 97. Please rephrase. Although your reputation on research on probiotics is not disputed, relevance should be clarified and the point of performing the research pointed out.
Methods: You state that you used standard curves, however these are not presented. Please provide equations and their correlation coefficients were appropriate (e.g. ascorbic acid, DPPH).
Methods
My major concern was whether you checked growth curves in the presence and absence of alginite. A reduced growth in the presence of alginite might clarify your results.
p4, L 166. In sample C you state that 400 μL of the appropriate amount of sample is used. Please modify since the volume (amount) is already stated. Also please state volume compensation solvent and total volume for each sample.
Results: Figures 3,4, 5 please state number of replicates and the bars (SD or SEM?) Please note that Bifidobacterium adolescentis is mentioned in two different ways. Please add bars in Fig. 5. All bacteria strains should be in italics.
Discussion seems incomplete and a repetition of results.
Conclusion Your statement regrading the ideal formula is presumptuous. Since no long term exposure data are unavailable or possible skin irritation effects it is not safe to draw such a conclusion.
Finally did you actually intend to use alginite or alginate?
Author Response
The authors are very grateful for receiving reviewers comments and suggestions to which they give the following answers:
This is a well presented work on the use of the culture free supernatant form probiotics grown in the presence of alginite on skin care products. However, the results were negative towards the intended target, which was moisturization, and the authors recommend their product for self tanning purposes. Such a product should be further evaluated for skin irritations on long term use.
Authors agree with Reviewer, that for application as self tanning agent the investigated “product” needs further tests including skin-irritation, that is the reason, while in the conclusion of the original manuscript this was also indicated. Thus, we kept it unchanged to highlight both the usefulness of the fermentation broth and the required further development direction.
I would also like to draw your attention to the following:
Introduction: The first sentence is too general. Your target is the skin and it might be appropriate to elaborate on the existing literature. Please provide relevant references.
Authors agreed with Reviewer, and removed the first general sentence therefore now the manuscript starts with the skin’s literature.
Introduction p 3, L 97. Please rephrase. Although your reputation on research on probiotics is not disputed, relevance should be clarified and the point of performing the research pointed out.
The requested sentence (now in line 102-104) was reformulated and our previous probiotic research become cited.
Methods: You state that you used standard curves, however these are not presented. Please provide equations and their correlation coefficients were appropriate (e.g. ascorbic acid, DPPH).
The “standard curves” expression remained in the text from the original methods, thus we modified the method descriptions according to our application. In case of DPPH measurement ascorbic acid as reference was used, and its curves parameters are implemented into the new Table 2. In case of tyrosinase measurement inhibition % could be determined without any calibration or standard curves. Koji acid was used for checking suitability of the tyrosinase measurement at different concentrations, but it was not used for samples evaluation, thus these results are not implemented.
Methods
My major concern was whether you checked growth curves in the presence and absence of alginite. A reduced growth in the presence of alginite might clarify your results.
Yes, we checked growth curves both in the presence and in the absence of alginite, but its presence enhanced the cell growth, which is now under publication elsewhere (Per Pol Chem Eng).
p4, L 166. In sample C you state that 400 μL of the appropriate amount of sample is used. Please modify since the volume (amount) is already stated. Also please state volume compensation solvent and total volume for each sample.
We are very grateful for the Reviewer detecting this mistake, which we corrected: all A,B,C,D measurement had 950microL volume +50microL tyrosinase, but unfortunately L-Dopa was missing from cuvette C, which is now involved in the manuscript.
Results: Figures 3,4, 5 please state number of replicates and the bars (SD or SEM?) Please note that Bifidobacterium adolescentis is mentioned in two different ways. Please add bars in Fig. 5. All bacteria strains should be in italics.
All measurements are done in triplicates, which is now indicated in the Materials and Methods descriptions. All Figures now have error bars indicating standard deviations (SD). Bacterial names were corrected.
Discussion seems incomplete and a repetition of results. Conclusion Your statement regrading the ideal formula is presumptuous. Since no long term exposure data are unavailable or possible skin irritation effects it is not safe to draw such a conclusion.
Results and Discussion are reformulated, thus the consideration of results are detailed in Discussion part.
Finally did you actually intend to use alginite or alginate?
Alginite was used in this whole work.
Round 2
Reviewer 1 Report
The author answered the all quires with proper explanations. Hence, I would recommend for the publish in the present form of the manuscript
Reviewer 3 Report
All points raised were addressed.